# Study of the Heat Exhaust Coefficient of Lateral Smoke Exhaust in Tunnel Fires: The Effect of Tunnel Width and Transverse Position of the Fire Source

Qiulin Liu 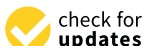, Zhisheng Xu, Weikun Xu, Sylvain Marcial Sakepa Tagne, Haowen Tao, Jiaming Zhao and Houlin Ying *

School of Civil Engineering, Central South University (CSU), Changsha 410075, China
* Correspondence: evelynlin@csu.edu.cn; Tel.: +86-731-8265-5177

**Abstract:** The tunnel width and the transverse fire's position both impact the heat exhaust coefficient, which is a critical component of the lateral smoke exhaust in tunnel fires. In this research, the tunnel width and the transverse location of the fire source are varied to analyze the heat exhaust coefficient of lateral smoke exhaust. When tunnel width increases, there is a noticeable decrease in the longitudinal temperature of the entrained air and smoke layer in the fire plume. Furthermore, the heat exhaust coefficients are reduced. An increase in the distance between the exhaust vent and the fire source causes an increase in the proportion of hot smoke in the smoke exhaust mass flow, which increases the heat exhaust coefficient. A calculated heat exhaust coefficient was developed using the fire source's location and the tunnel's width as inputs, which agrees well with the simulation results. This method can predict the heat exhaust coefficient of the lateral smoke exhaust in tunnel fires. The findings of this study provide insight into how the tunnel width and the location of a transverse fire influence the heat exhaust coefficient.

**Keywords:** tunnel fire; heat exhaust coefficient; tunnel width; transverse fire location; lateral smoke exhaust

## 1. Introduction

The development of construction technology has been flourishing, and the number of tunnels with large diameter increases continually [1]. However, the studies on fire safety in tunnels with large diameters are still insufficient [2]. Since the heat dissipation is restricted and the hot smoke spreads longitudinally along the tunnel during tunnel fires [3,4], using a reasonable ventilation system to prevent the spread of smoke and harmful gases is crucial [5,6]. The lateral smoke exhaust has been adopted in various tunnels [7]. Nevertheless, the change in the tunnel width [8,9] and the transverse position of the fire source [10,11] influence the heat exhaust coefficient.

According to Chen et al.'s study [12], the maximum temperature rise of the fire and the attenuation coefficient of smoke flow influenced the heat lost by the smoke exhaust. However, the tunnel width influenced the maximum temperature rise and the attenuation factor of smoke close to the fire source. [13]. Additionally, this investigation did not consider the amount of smoke in the expelled mass flow. The amount of smoke in the exhaust mass flow caused by air entrainment at the exhaust vent is affected by the plug-holing phenomena. The plug-holing phenomenon would also be experienced by the lateral smoke exhaust, according to Yang et al. [14]. When plug-holing occurs, a great deal of cold air is entrained, which reduces the efficiency of smoke extraction. Jiang et al. [15] calculated the smoke back-layering length in their study. They redefined the confinement velocity based on the smoke temperature distribution below the tunnel ceiling. Zhong et al. [16] proposed the concept of a virtual smoke vent based on the results of different smoke exhaust



velocities, heat release rates and smoke layer heights in their study of the occurrence of plug-holing penetration in the lateral smoke exhaust. A critical Froude number for predicting lateral smoke exhaust plug-holing was established and a corresponding critical Froude number of 0.5 was given for the occurrence of plug-holing. Zhu et al. [17] proposed a modified Froude number for determining the plug-holing phenomenon through a tunnel fire under the action of lateral smoke exhaust. The critical Froude number ranges from approximately 1.5 to 1.75. Our previous study [18] considered the difference between lateral smoke exhaust and ceiling smoke exhaust. We modified the Froude number to determine the plug-holing of lateral smoke exhaust to establish a new modified Froude number. The results show that when the new modified Froude number is greater than 2.5, the lateral smoke exhaust experiences plug-holing.

However, previous research has concentrated on determining plug-holing in the lateral smoke exhaust. There is a lack of research into the effect of different tunnel widths and the lateral position of the fire source on the effectiveness of lateral smoke exhaust. Additionally, the direction of extraction is different for the ceiling smoke exhaust and the lateral smoke exhaust, especially in large-diameter tunnels. The cold air is encased by the smoke layer and exhaust vent working together. Meanwhile, there is a lack of correlation between the tunnel width, the transverse fire location, and the heat exhaust efficiency. Therefore, by large eddy simulation (LES) and theoretical analysis, the change of the influencing factors heat exhaust coefficient in tunnel fires with various tunnel widths and transverse fire locations was studied.

## 2. Theoretical Analysis

The heat exhaust coefficient determines the quantity of heat expelled [19], which is the ratio of heat exhausted to heat dispersed from the fire source to one side of the tunnel.

$$\eta = \frac{\dot{Q}_{es}}{C_x \dot{Q}_c} = \frac{c_p \dot{m}_s \Delta T_r}{C_x \dot{Q}_c} = \frac{c_p \varphi_s \dot{m}_{es} \Delta T_r}{C_x \dot{Q}_c} \tag{1}$$

where:
$\dot{Q}_{es}$: heat carried by the smoke discharged from exhaust vent (kW);
$C_x$: correction factor;
$\dot{Q}_c$: convective heat release rate (kW);
$c_p$: thermal capacity of air (kJ/(kg·K));
$\dot{m}_s$: mass flow of the hot smoke exhausted by the exhaust vent (kg/s);
$\varphi_s$: proportion of smoke in the exhausted mass flow;
$\dot{m}_{es}$ mass flow of the exhausted gas (kg/s);
$\Delta T_r$: temperature increase in the smoke layer at the exhaust vent (K).

According to the study of Ingason et al. [20], radiative heat constitutes between 20 and 40 percent of total heat release, as a result, $\dot{Q}_c$ can be described as:

$$\dot{Q}_c = 0.7\dot{Q} \tag{2}$$

where:
$\dot{Q}$: total heat release rate of the fire source (kW).

For a particular fire scenario, $\dot{Q}_c$ and $\dot{m}_{es}$ are almost constant. According to Equation (1), it can be inferred that the heat exhaust coefficient mainly relates to $\varphi_s$ and $\Delta T_r$.

As shown in Figure 1, the flow field beside the exhaust vent becomes a disturbance, and hot smoke is mixed with cold air more intensely during the exhausting process. $\dot{m}_{a1}$ is a representation of the mass flow of the cold air directly discharged by the exhaust vent, $\dot{m}_{a2}$ is a representation of the mass flow of the cold air entrained by the smoke layer.

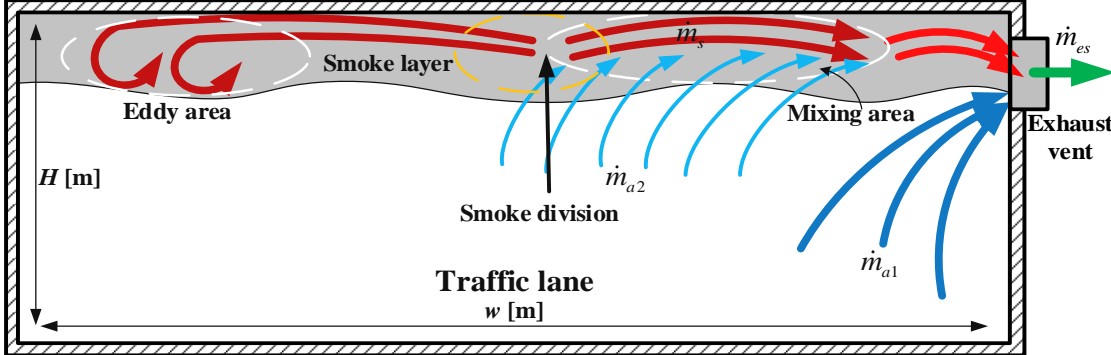

**Figure 1.** Schematic diagram of the flow field beside the exhaust vent.

According to the principle of conservation of mass, it can be obtained that:

$$\dot{m}_{es} = \dot{m}_s + \dot{m}_{a1} + \dot{m}_{a2} \tag{3}$$

$\varphi_{a1}$ is the proportion of $\dot{m}_{a1}$ in the exhausted mass flow:

$$\varphi_{a1} = \frac{\dot{m}_{a1}}{\dot{m}_{es}} \tag{4}$$

In the early research, Hinkley [21] analyzed the plug-holing phenomenon, which was prompted by natural smoke extraction. Hinkley proposed a dimensionless number *Fr* to show the plug-holing that took place, which can be expressed as Equation (5).

$$Fr = \frac{u_v A}{(g\Delta T/T_0)^{1/2} d^{5/2}} \tag{5}$$

where:
$g$: gravitational acceleration (m/s$^2$);
$\Delta T$: temperature rise of the smoke layer (K);
$d$: thickness of the smoke layer (m);
$u_v$: flow velocity at the exhaust vent (m/s);
$A$: area of the exhaust vent (m$^2$).

The study of Ji et al. [22] has shown that the air entrainment in the exhausted mass flow could increase to 48%, where no apparent plug-holing phenomenon is observed. The *Fr* number might be intimately connected to the exhaust mass flow's air entrainment. Furthermore, according to Cong et al. [23], the *Fr* value may be used to calculate the air entrainment at the exhaust vent. So, the values of *Fr* are related to the proportion of $\dot{m}_{a1}$ in $\dot{m}_{es}$, i.e.,

$$\varphi_{a1} = C_f Fr = \frac{C_f u_v A}{(g\Delta T/T_0)^{1/2} d^{5/2}} \tag{6}$$

where:
$C_f$: correction factor.

As shown in Figure 1, the division phenomenon of the smoke layer occurs under the dual function of the eddy area and the smoke exhaust vent, and cold air $\dot{m}_{a2}$ is sucked into the smoke layer, $\varphi_{a2}$ is a representation of the ratio of $\dot{m}_{a2}$ to the mass flow of the exhausted gas ($\dot{m}_{es}$):

$$\varphi_{a2} = \frac{\dot{m}_{a2}}{\dot{m}_{es}} \tag{7}$$

According to Equation (3), the percentage of smoke in the exhausted mass flow ($\varphi_s$), can be expressed as:

$$\varphi_s = 1 - \varphi_{a1} - \varphi_{a2} = 1 - \frac{C_f u_v A}{(g\Delta T/T_0)^{1/2} d^{5/2}} - \frac{\dot{m}_{a2}}{\dot{m}_{es}} \tag{8}$$

In the FDS simulation, $\Delta T$ can be measured by placing thermocouples; $u_v$ can be measured by placing a "Velocity Device" at the exhaust vent; $d$ can be measured by setting up a "Layer Zoning Device"; $\dot{m}_{es}$ can be measured by placing a "Mass Flow Device" at the exhaust vent; $\dot{m}_s$ can be measured at the cross-section of the tunnel with the "Mass Flow Device". The parameters measured in different scenarios are substituted into Equation (8) and then the remaining unknown parameters are obtained by associating Equation (8) with different scenarios.

## 3. Numerical Modeling

FDS has been extensively applied in fire and smoke control research and has been widely verified for its reliability. The Large Eddy Simulation (LES), which uses a filter function to separate the large eddy from the small eddy, is used in FDS to solve the Reynolds stress. Numerical computations are used to unravel the large eddy, while Sub-Grid Models (SGM) are used to unravel the small eddy.

### 3.1. Model Design

As illustrated in Figure 2, the tunnel model used in this study is a full-size tunnel model of 200 m long, 7 m high and 10/15/20/25 m wide. The wall material of the tunnel in this paper is concrete, with a friction factor set to 0.022. In addition to a Prandtl number of FDS default value of 0.5. The Stanton number was calculated from the above parameters to be 0.017. The fire source is situated at varied transverse distances from the tunnel side wall and 50 m from the exit at the left end of the tunnel. In addition, the fire source is 0.5 m from the ground. The lateral smoke exhaust vent of 4 m × 1.2 m is positioned 80 m from the exit at the left end of the tunnel.

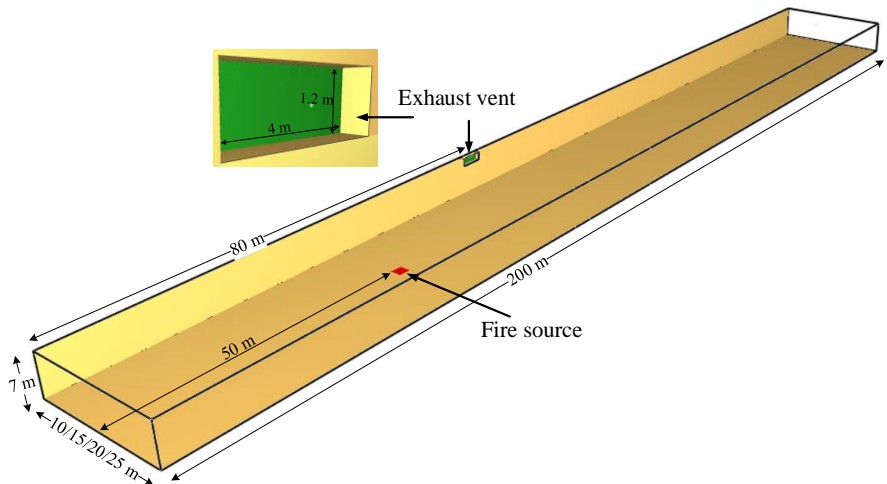

**Figure 2.** Model configuration.

Numerous studies [24–26] have found the most common fire scenarios to be 3–5 MW for car fires and 10–20 MW for truck fires in road tunnels. So, the heat release rate is set at 12 MW in this paper. There was a 293 K ambient temperature and a 101 kPa ambient pressure set. The border conditions of the tunnel's two portals were designated as "OPEN". A 400 s simulation was conducted. The grid size we chose is 0.25 m to simulate the tunnel fire.

As demonstrated in Figure 3, the thermocouples positioned 0.2 m below the ceiling centerline and spaced 1 m were developed to detect the temperature distribution at the ceiling. Temperatures of the smoke were measured using thermocouples positioned vertically, placed 15 m from the fire source, and spaced 0.2 m. The "Layer Zoning Device" in FDS was used to gauge the smoke layer where the exhaust vent is located, which can measure the thickness of the smoke layer. A "Mass Flow Device" and a "Heat Flow Device" was mounted inside the smoke layer to measure the amount of heat transported by the smoke layer per unit mass at the site of the exhaust vent. Meanwhile, the exhaust vent's heat and mass flow were detected by the function of the "Mass Flow Device" and "Heat Flow Device" placed in front of the exhaust vent. A velocity measurement was made at the exhaust vent using the "Velocity Device" in FDS.

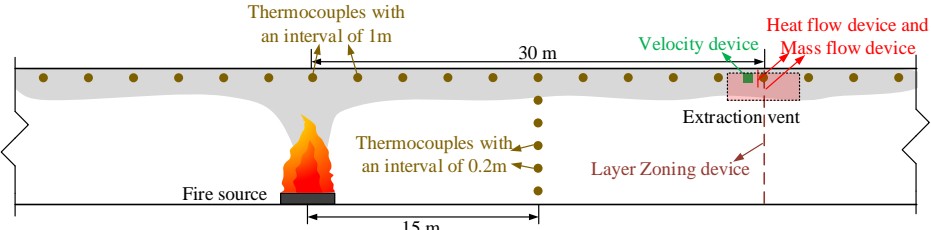

**Figure 3.** Probe arrangement.

A set of temperature and velocity slices were set up at the smoke vent and the lateral position of the fire source to obtain the temperature and velocity flow fields at the fire source and smoke vent. The slices were processed with Tecplot software to obtain the temperature and velocity flow fields. The average value of the stable section from 300 s to 400 s was extracted from the simulation results for discussion.

### 3.2. Simulated Conditions

Table 1 shows the parameters under every 32 scenarios in this research. In this study, two mass flow rates ($\dot{m}_{es}$) of 0 kg/s (no smoke exhaust) and 12 kg/s (smoke exhaust) were determined for the smoke exhaust vent. The width of the tunnel increased from 10 m adding 5 m in sequence up to 25 m. The fire sources were in different transverse locations of the tunnel.

**Table 1.** Summary of the scenarios.

| NO. | HRR (MW) | Mass Flow Rate of the Exhaust Vent (kg/s) | Width of the Tunnel (m) | Transverse Distance between Fire Source and Side Wall (m) |
|---|---|---|---|---|
| 1–3 | | | 10 | 2.5, 5, 7.5 |
| 4–6 | | | 15 | 2.5, 7.5, 12.5 |
| 7–11 | | 0 | 20 | 2.5, 7.5, 10, 12.5, 17.5 |
| 12–16 | 12 | | 25 | 2.5,7.5, 12.5, 17.5, 22.5 |
| 17–19 | | | 10 | 2.5, 5, 7.5 |
| 20–22 | | | 15 | 2.5, 7.5, 12.5 |
| 23–27 | | 12 | 20 | 2.5, 7.5, 10, 12.5, 17.5 |
| 28–32 | | | 25 | 2.5,7.5, 12.5, 17.5, 22.5 |

We calculated the maximum temperature rise below the tunnel ceiling (tunnel width: 25 m; transverse distance between fire source and side wall: 2/3/4/5/6 m) and compared it with the findings of the model-scale test of Ji et al. [27] (Equation (9)), as shown in Figure 4. The results of the numerical simulations are in general agreement with the results of the model-scale test. However, there are still some differences between the two results. The reasons for the deviation between the two results are complex. Firstly, due to the tightness of the assumptions made in the numerical simulation model, there

is some influence on the maximum temperature rise below the tunnel ceiling. In their experiments, Ji et al. did not specify any parameters for the tunnel wall material. As a result, the materials used in Ji et al.'s model-scale test have different specific heat values, densities, and thermal conductivities than the concrete described in this work. Which may also alter the maximum temperature rise under the tunnel ceiling. However, the results of the numerical simulations in this paper agree with the data from the previous model-scale test. The numerical simulations can therefore be used to investigate the fire source's smoke flow at different widths and lateral positions.

$$\Delta T_{\text{max},D} = 17.9 \frac{\dot{Q}^{2/3}}{H^{5/3}} (1.096 e^{-14.078D/(w/2)} + 1) \tag{9}$$

where:
$\Delta T_{\text{max},D}$: maximum temperature rise at a distance $D$ from the side wall (K);
$H$: tunnel height (m);
$D$: distance from the side wall (m);
$w$: tunnel width (m).

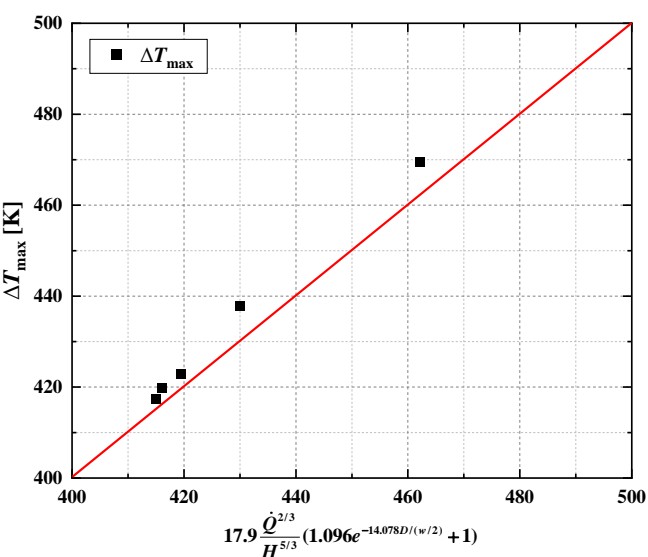

**Figure 4.** Comparison of the maximum temperature rise at different locations with Ji's model (Equation (9)).

## 4. Results and Discussion

The variation of the factors influencing the heat exhaust coefficient was studied by varying two variables: the fire source's transverse position and the tunnel's width. A model for predicting the heat exhaust coefficient in tunnel fires was proposed.

### 4.1. Effect of Tunnel Width on Fire Plume without Smoke Exhaust

Since tunnel width increases with tunnel diameter, there is a change in both the transverse spread distance of smoke and the cold air entrained by the fire plume. Figure 5 demonstrates the development of fire plumes in tunnels with different widths. When the tunnel width is 10 m and 15 m, the fire plume's air entrainment in the tunnel's transverse direction is less than when the tunnel is wider. Most of the air between the tunnel side wall and the fire source gets heated significantly when the tunnel width is 10 m. Air entrainment by the fire plume behaves similarly as tunnel width increases, decreasing smoke temperatures. In the diffuse stage of smoke flow, when the smoke flow diffuses to the tunnel sidewall, an anti-buoyancy wall jet is formed by the smoke flow moving vertically downward alongside the sidewall, resulting in the backflow of some smoke. As the width of the tunnel increases, the greater the resistance in the process of diffusion of

smoke to the tunnel sidewalls, resulting in a gradual weakening of the smoke's horizontal inertia force. Therefore, the wider the tunnel, the weaker the anti-buoyancy wall jet, and its vertical length decreases significantly. The smoke layer's thickness also tends to thin as tunnel width increases. The wider the tunnel, the thinner the smoke layer will be under the same smoke production volume.

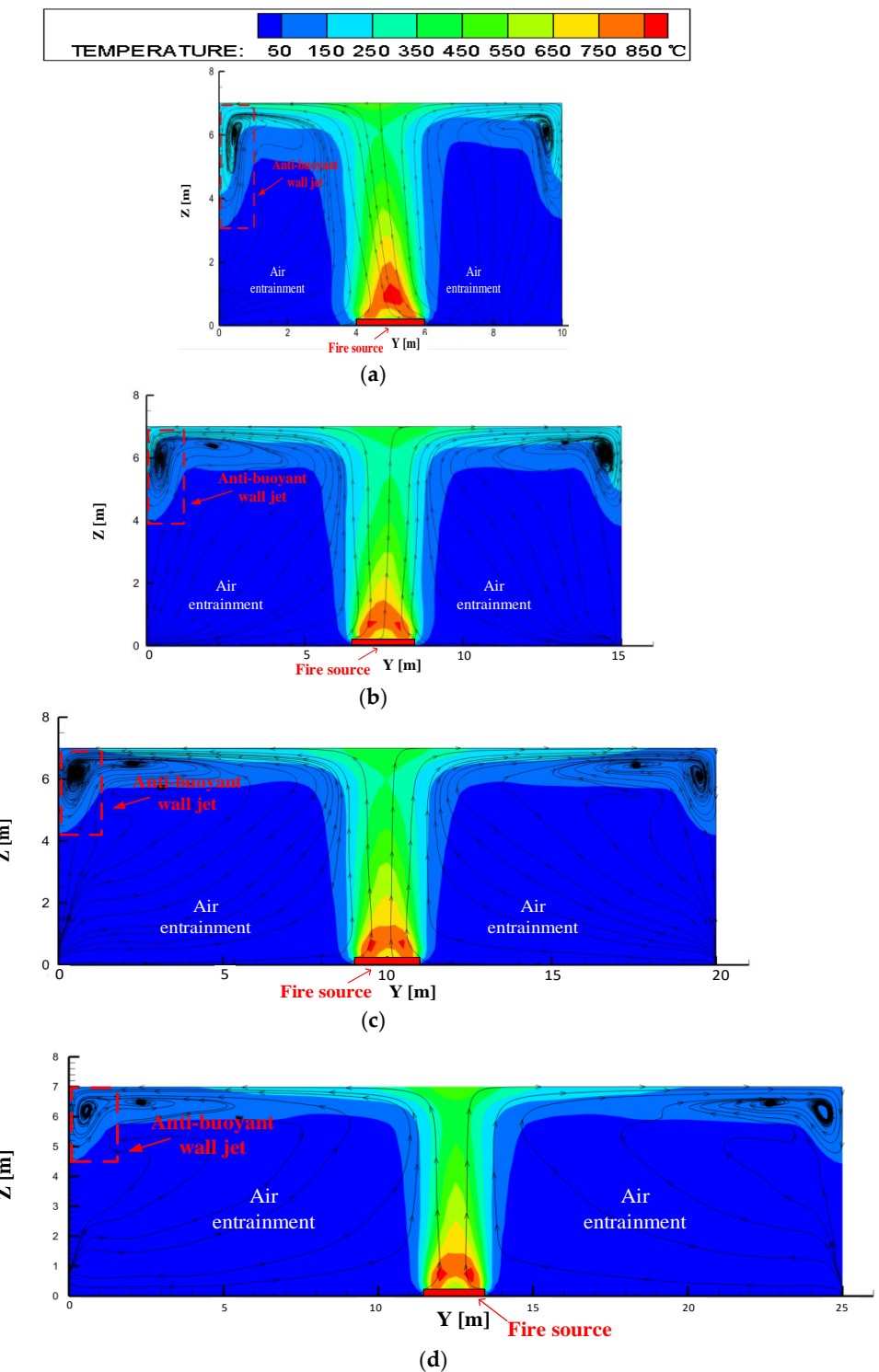

**Figure 5.** Temperature and velocity flow distribution around the fire source for different tunnel widths: (**a**) 10 m; (**b**) 15 m; (**c**) 20 m; (**d**) 25 m.

### 4.2. Effect of Fire Source Transverse Location on Heat Exhaust Coefficient

　　As the smoke from the fire source hits the side walls under the effect of the transverse inertial force, as depicted in Figure 6, an eddy area forms under the ceiling. The eddy area under the ceiling on the side farther from the fire source becomes larger. The size of the vortex area affects the heat exhaust coefficient. The eddy area also spreads forward during the longitudinal spread of smoke, affecting lateral smoke exhaust. As depicted in Figure 7, when the fire source is situated at distances of 2.5 m and 7.5 m, there is a large range of strong eddy areas under the ceiling far from the exhaust vent, resulting in the division of hot smoke. A mass of cold air enters the smoke layer and mixes with the hot smoke, ultimately decreasing the heat exhaust coefficient. When the fire source is positioned at the center of 12.5 m, then the range of the strong eddy area is reduced. Although the hot smoke diversion phenomenon still occurs, the position of smoke diversion is farther away from the exhaust vent. The extraction exhausts more hot smoke, improving the heat exhaust coefficient. If the fire source is positioned at 17.5 m and 22.5 m, there is no obvious eddy area and smoke diversion phenomenon in the smoke layer. Most of the hot smoke moves toward the exhaust vent, effectively reducing the mixing of cold air and significantly improving the heat exhaust coefficient.

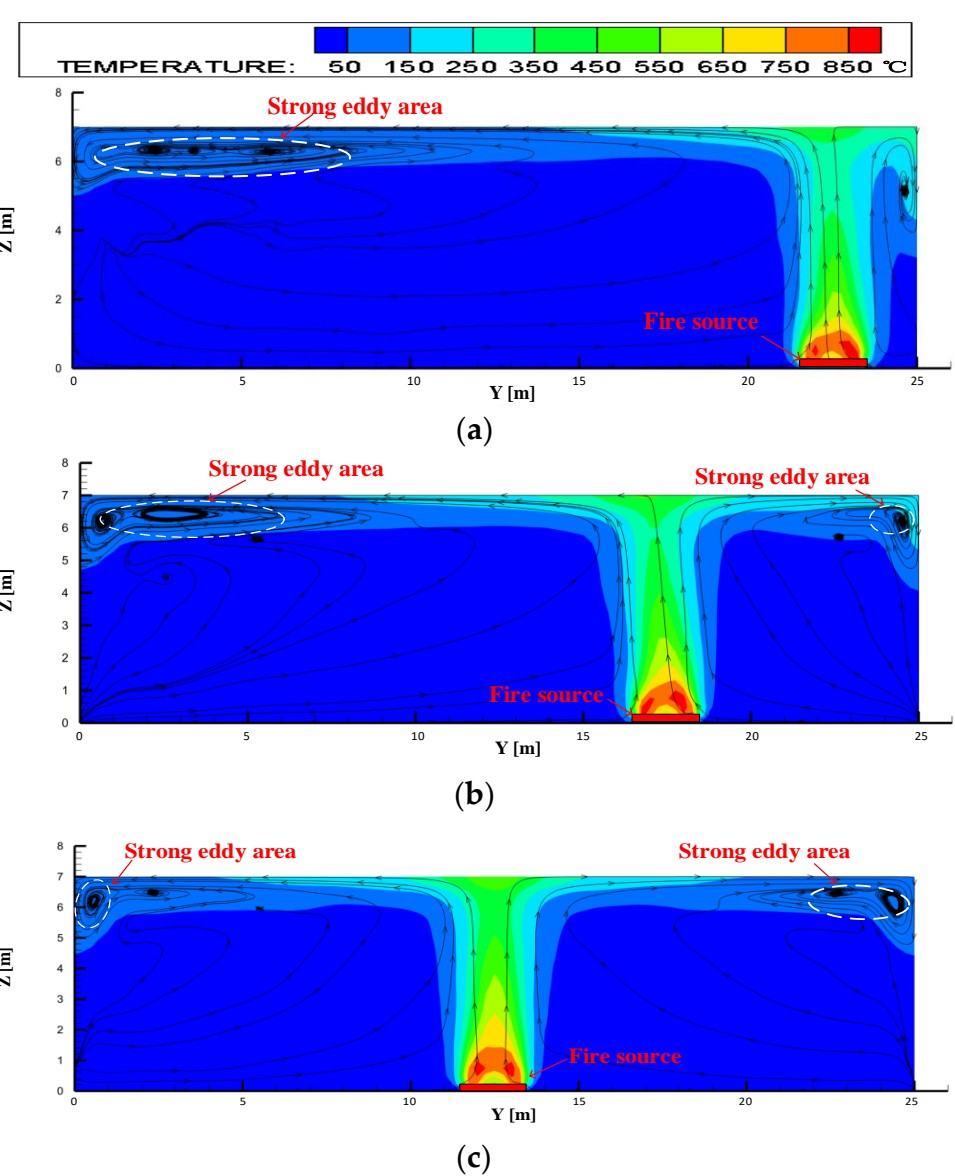

**Figure 6.** *Cont.*

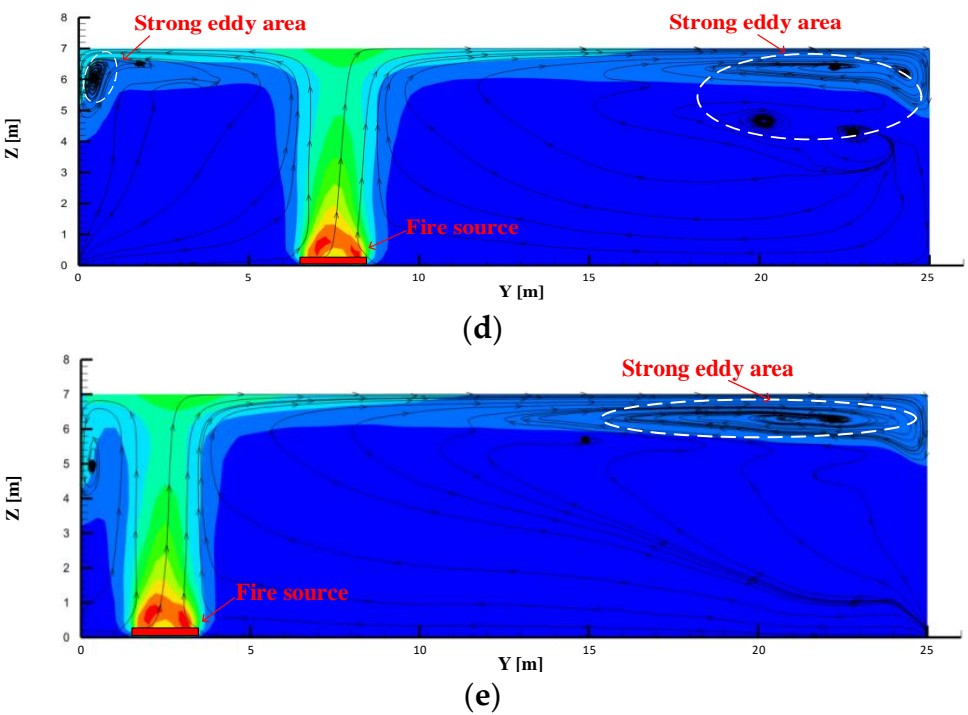

**Figure 6.** Temperature and velocity flow distribution around the fire source in the 25 m wide tunnel for different distances from the side walls: (**a**) 2.5 m; (**b**) 7.5 m; (**c**) 12.5 m; (**d**) 17.5 m; (**e**) 22.5 m.

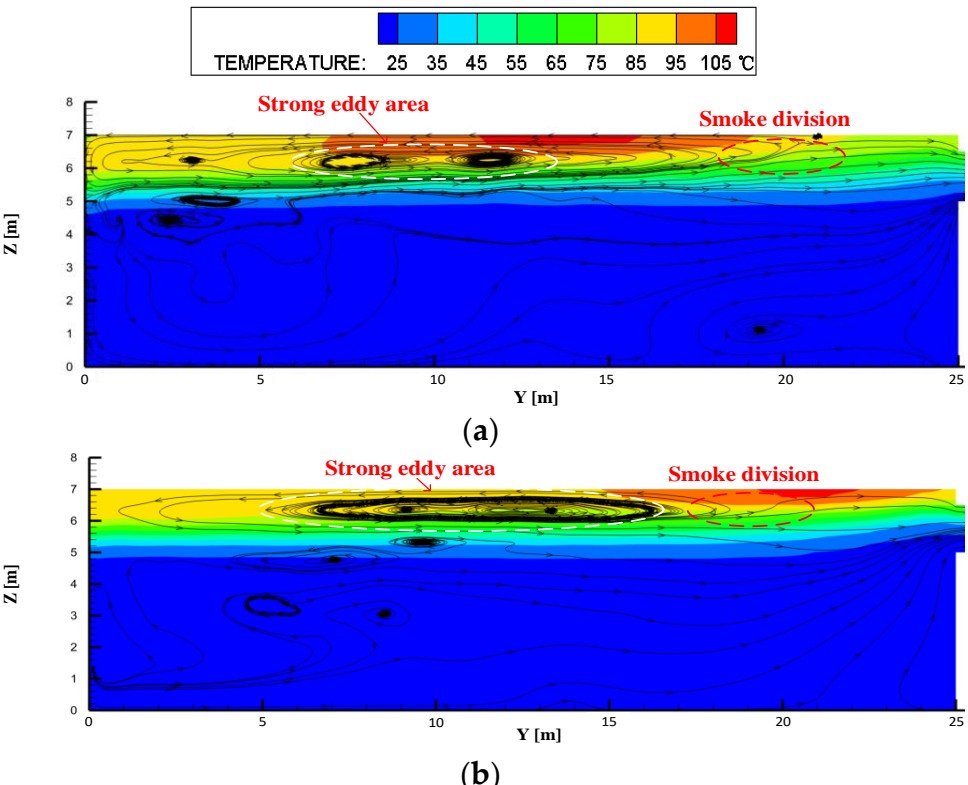

**Figure 7.** *Cont.*

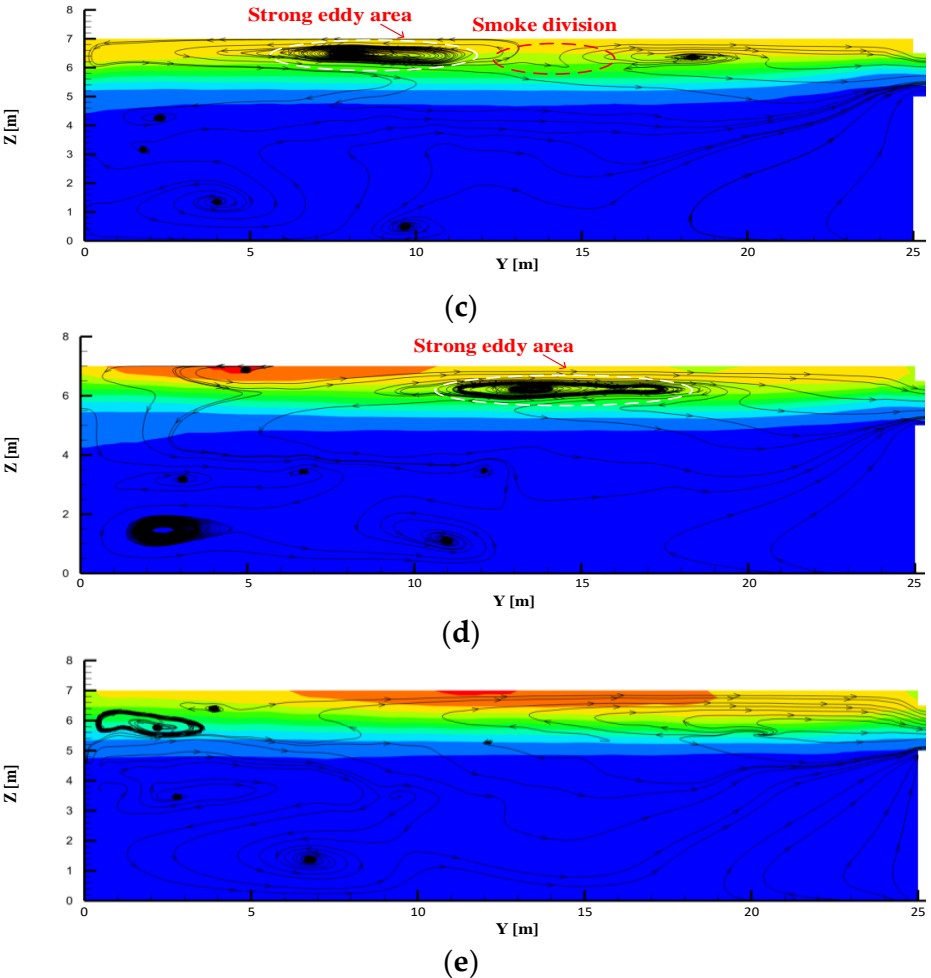

**Figure 7.** Temperature and velocity flow distribution around the exhaust vent in the 25 m wide tunnel for different distances from the side walls: (**a**) 2.5 m; (**b**) 7.5 m; (**c**) 12.5 m; (**d**) 17.5 m; (**e**) 22.5 m.

It can be deferred that the ratio of the transverse fire source distance from the exhaust vent ($L$) to the tunnel width ($w$) will affect the $\varphi_{a2}$ by observing Figures 6 and 7. So, $\varphi_{a2}$ can be described as:

$$\varphi_{a2} = f(\frac{L}{w}) \tag{10}$$

Figures 8 and 9 show the correlation between the proportion of hot smoke and cool air in the smoke exhausted by the lateral exhaust vent in the tunnel that is 20 m and 25 m wide. When there is an increase in the transverse distance between the lateral exhaust vent and the fire source, the proportion of hot smoke increases while the proportion of cold air drops.

In the 20 m wide tunnel, the following formula can be used to represent the percentage of cold air in the exhausted mass flow of the lateral exhaust vent:

$$\varphi_{a1} + \varphi_{a2} = -0.088 \ln(\frac{L}{w}) + 0.5156 \tag{11}$$

In the 25 m wide tunnel, the following formula can be used to represent the percentage of cold air in the exhausted mass flow of the lateral exhaust vent:

$$\varphi_{a1} + \varphi_{a2} = -0.088 \ln(\frac{L}{w}) + 0.5361 \tag{12}$$

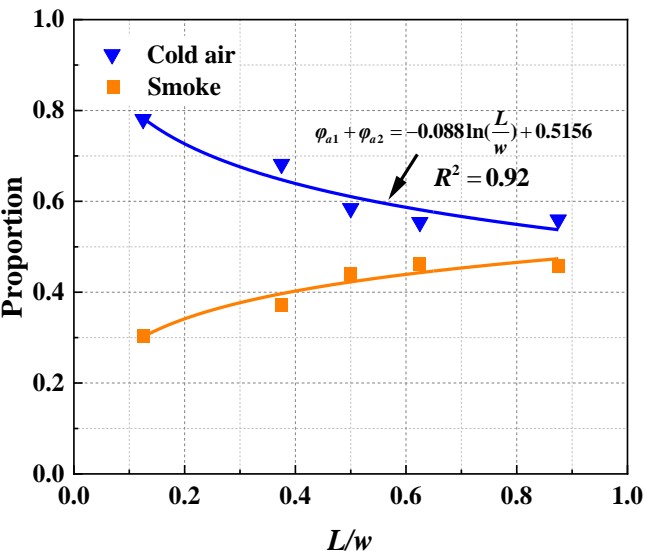

**Figure 8.** The proportion of hot smoke and cold air in the exhaust gas in a 20 m wide tunnel.

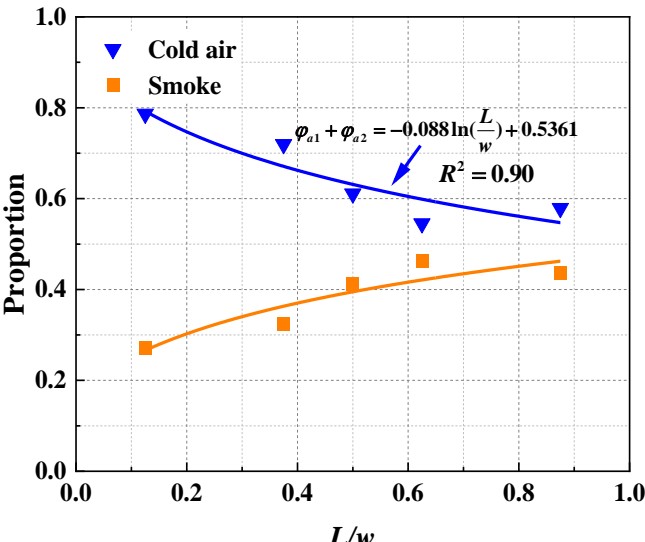

**Figure 9.** The proportion of hot smoke and cold air in the exhaust gas in a 25 m wide tunnel.

The value of $Fr$ is 2.04 for a tunnel with a width of 20 m and 2.41 for a tunnel with a width of 25 m. When Equation (6) is combined with Equations (10)–(12), the following results are produced:

$$\begin{cases} 2.04C_f + f(\frac{L}{w}) = -0.088\ln(\frac{L}{w}) + 0.5156 \\ 2.41C_f + f(\frac{L}{w}) = -0.088\ln(\frac{L}{w}) + 0.5361 \end{cases} \tag{13}$$

Then, the value of $C_f$ can be obtained as 0.056. So $\varphi_{a1}$ can be expressed as:

$$\varphi_{a1} = 0.056Fr \tag{14}$$

Substituting Equation (14) into Equation (12), $\varphi_{a2}$ can be expressed as:

$$\varphi_{a2} = -0.088\ln(\frac{L}{w}) + 0.40 \tag{15}$$

Substituting Equations (14) and (15) into Equation (8), $\varphi_s$ can be expressed as:

$$\varphi_s = 0.6 + 0.088\ln(\frac{L}{w}) - 0.056Fr \qquad (16)$$

The amount of hot smoke in the expelled mass flow grows as the transverse distance between the lateral exhaust vent and the fire source increases, ultimately leading to a larger heat exhaust coefficient.

*4.3. Verification of the Effect of Lateral Extraction Efficiency*

The value of $C_x$ (Equation (1)) can be obtained to be 0.44, combining the simulation results obtained from scenarios 1~32. Then the heat exhaust coefficient can be described as:

$$\begin{cases} \eta = \dfrac{\Delta T_r \varphi_s \dot{m}_{es}}{0.44\dot{Q}_c} \\ \varphi_s = 0.6 + 0.088\ln(\frac{L}{w}) - 0.056Fr \end{cases} \qquad (17)$$

As shown in Figure 10, Equation (17) matches the simulation results quite well. It is clear in Figure 10 that the heat exhaust coefficient increases with the transverse interval from the fire to the exhaust vent, while it declines with the growth of the tunnel width.

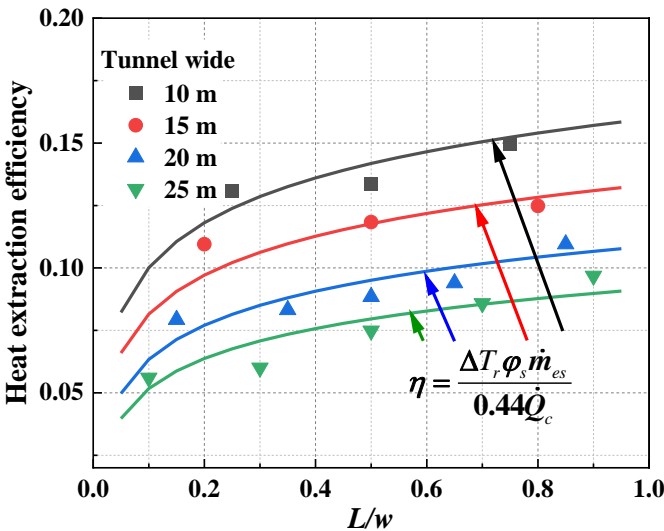

**Figure 10.** Comparison of Equation (17) with numerical simulation results.

**5. Conclusions**

This paper used a series of LES methods to conduct a simulation. The heat exhaust coefficient associated with tunnel fire lateral smoke exhaust has been investigated. A correlation for predicting the heat exhaust coefficient has been proposed. The main conclusions are:

(1) There will be an increase in the amount of hot smoke in the exhaust mass flow if the transverse distance between the exhaust vent and the fire source increases, increasing the heat exhaust coefficient in tunnel fires.

(2) The more expansive the tunnel, the weaker the anti-buoyancy wall jet, and its vertical length decreases significantly. In addition, as the tunnel width grows, the smoke layer thickness becomes gradually thinner.

(3) A calculation model was developed to compute the heat exhaust coefficient (Figure 10, Equation (17)) taking into account the lateral position of the fire source and the width of the tunnel.

**Author Contributions:** Methodology, Z.X.; software, W.X.; validation, H.T.; formal analysis, Q.L.; investigation, Q.L.; resources, H.Y.; data curation, J.Z.; writing—original draft preparation, Q.L.; writing—review and editing, S.M.S.T. All authors have read and agreed to the published version of the manuscript.

**Funding:** The Fundamental Research Funds for the Central Universities of Central South University (No. 2021zzts235) provided funding for this project.

**Institutional Review Board Statement:** Not applicable.

**Informed Consent Statement:** Not applicable.

**Data Availability Statement:** Not applicable.

**Acknowledgments:** We acknowledge the High Performance Computing Center of Central South University for its support. This work was supported by the Natural Science Foundation of Hunan Province of China (Grant No. 2020JJ3046) and the Fundamental Research Funds for the Central Universities of Central South University (Grant No. 2021zzts235).

**Conflicts of Interest:** The authors declare no conflict of interest.

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
