# Peer review of "Study of the Heat Exhaust Coefficient of Lateral Smoke Exhaust in Tunnel Fires: The Effect of Tunnel Width and Transverse Position of the Fire Source"

_fire, doi:10.3390/fire5050167_

Round 1
Reviewer 1 Report
This paper conducted the numerical simulation on the effects of tunnel width and the transverse location of the fire source on the heat exhaust coefficient for the tunnel fire by lateral smoke exhaust system., a calculation model was developed in order to compute the heat exhaust coefficient by taking the lateral position of the fire source and the width of the tunnel into account. The findings of this study could provide insight on the lateral smoke exhaust heat exhaust system design for tunnel fire. The authors should revise the following points in order to better express the research results.
1. The background to the study of lateral exhaust system is not sufficiently described in the introduction, which should emphasize the practical nature of the study.
2. In subsection 3.1 “the tunnel model used in this study is a full-size tunnel 107 model of 200 meters long, 7 meters high and 10/15/20/25 meters wide”, please explain the basis for the full-size tunnel.
3. The values of Pr and St in the FDS simulation should be clearly stated.
4. This paper lacks verification of the validity of numerical simulations, and a subsection should be included in section3 for discussion.
5. Figures 1 and 7 are lost, please check the manuscript carefully.
6. There are some formatting errors in English, such as line103 “Solve”. Please check it carefully.
7. The English level in this article is poor, and some of the phrases are long and cumbersome. It is recommended to revise them helped by native English speakers.
Reviewer 2 Report
“Introduction” should not contain any mathematical models like equation (1). They can be easily moved to the part called “Theoretical analysis”. Include into “Introduction” clear statements about the gap of knowledge covered by the manuscript and added value proposed by the study. Both must be substantiated by extensive citations, because the body of theoretical work in the topic under analysis is large. Currently, the manuscript seems to be a simple collection of formulas called “Theoretical analysis” and ordinary PyroSim+FDS simulation typical for standard industrial applications. The simulation part is detached from “Theoretical analysis”. It is not explained in detail how all the theoretical business was implemented in the use of FDS code. Without such an explanation, “Theoretical analysis” looks like a wastage of journal space.
The statement “A 12 MW heat release rate was determined in this study for the most ordinary car and truck fire scenes in road tunnels” is wrong as is the reliance on the value of 12 MW. First, it is not clear what are “ordinary” cars if we look at a huge variety of vehicles on roads. Second, HRRs of trucks depend heavily on the load and dimensions of trailers and tractors. This can be easily illustrated by the Swedish experiments on fires in Norwegian tunnels (experiments of H. Ingason and his co-authors). As the choice of HRR for the vehicle fire in tunnel determines all subsequent results, the HRR model must be chosen and substantiated with maximum care. Modelling must be redone correspondingly. The currently proposed simulation is good for practice but insufficient for scientific research. My master students do similar simulations on daily basis in small private companies. Alternatively, authors can explicitly recognise that the on-floor circular pool fire is an ill-natured idealisation and serves only for the exploration of fire processes in the tunnel configuration under study.
The above comments on the fire source devaluate the section “Grid sensitivity analysis”. A truck consisting of tractor and trailer does not burn as a pool with a diameter of D meters. The same applies to other vehicles, to a lesser of greater degree. In addition, it is clear without any analysis that the grid geometry will influence simulation results. Study related to the FDS grid is too shallow for making the statement that 0.25 meters allow to get “reliable calculation results”. Proving this “reliability” is an individual and, in my opinion, intricate problem lying outside the problematic of tunnel fires. A simple statement that all simulation results are conditioned upon the 0.25 meters cell will be sufficient.
The authorship of each equation used in the manuscript must be indicated (cf, eg, Eq(2)).
The computer code used for a visualisation of model and modelling results in Figs. 2, 5 and 6 must be indicated explicitly. FDS code does not allow any visualisation without pre-processors and post-processors.
Round 2
Reviewer 1 Report
The authors carefully revised paper according to the comments of reviwers. I recommend that this paper can be accepted